# Beneš Network-Based Efficient Data Concentrator for Triggerless Data Acquisition Systems

Marek Gumiński [1,*], Michał Kruszewski [1], Bartosz Marek Zabołotny [2] and Wojciech Marek Zabołotny [1,*]

1   Institute of Electronic Systems, Faculty of Electronics and Information Technology,
    Warsaw University of Technology, Nowowiejska 15/19, 00-650 Warszawa, Poland
2   Institute of Telecommunications, Faculty of Electronics and Information Technology,
    Warsaw University of Technology, Nowowiejska 15/19, 00-650 Warszawa, Poland
*   Correspondence: marek.guminski@pw.edu.pl (M.G.); wojciech.zabolotny@pw.edu.pl (W.M.Z.)

**Abstract:** The concentration of data from multiple links to a single output is an essential task performed by High-Energy Physics (HEP) Data Acquisition Systems (DAQs). At high and varying data rates combined with the large width of the concentrator's output interface, this task is non-trivial. A high-speed dense packing of data from possibly non-continuous streams with preserving their time order requires complex and real-time adjustable routing. This paper presents a concentrator based on the Beneš network, which provides efficient concentration without using a high-frequency clock internally. It warrants that empty data are eliminated and does not disturb the data time-ordering if the data rates significantly differ between inputs. The concentrator uses simple data-routing primitives resulting in low resource consumption. If necessary, the pipeline registers may be added after each routing stage, shortening the critical path and increasing the maximum acceptable clock frequency. These features render the design well-suited to FPGA implementation.

**Keywords:** FPGA; DAQ; data concentration; Beneš network

## 1. Introduction

Most detectors in high-energy physics (HEP) experiments deliver massive data streams in multiple channels. Reception of this data and its delivery to the analyzing computers is the task of the readout chains. In the case of experiments using a trigger, the data must be processed locally to elaborate the level one (L1) trigger decision. Preparing it for concentration may be a side effect of this process. The data are zero suppressed—invalid or empty data words are removed from the stream. Finally, the data are buffered in memory, which may offer the data width conversion. It may be written with single data words and read with multiple words in parallel, as required by the DAQ interface.

In the triggerless readout, the situation is different. The readout system does not need to perform complex local processing of data. Extraction of interesting events is carried out in further stages of the DAQ ("event builder" and "event filter" [1] or "event selector" [2]). The responsibility of the readout system is different in this configuration. It should almost transparently transfer the detector data to DAQ. Therefore, a popular triggerless readout architecture is where the data streams from detector front-end boards are delivered to the DAQ computers without modification. Usually, the streams are only multiplexed for efficient transport via high-speed optical links (e.g., using the GBTX ASIC and GBT-FPGA core [3]). The reception of those streams and data concentration is performed by FPGA-based PCIe cards hosted in DAQ entry nodes. The PCI Express blocks in FPGA require specific data bus width in the AXI interface. Table 1 shows the available AXI data widths depending on the speed of the link, the width of the PCIe lane and the AXI clock frequency. For PCIe 8xGen3, it is necessary to work with 256-bits wide data.

**Table 1.** Width and clock frequency of AXI interface for PCI Express blocks. Results obtained from various configurations of AMD/Xilinx DMA/Bridge Subsystem for PCIe Express (4.1).

| Lane Width | Maximum Link Speed | | |
| --- | --- | --- | --- |
| | 2.5 GT/s (Gen 1) | 5 GT/s (Gen 2) | 8 GT/s (Gen 3) |
| 1 | 64 bits @ 62.5 MHz<br>64 bits @ 125 MHz<br>64 bits @ 250 MHz | 64 bits @ 62.5 MHz<br>64 bits @ 125 MHz<br>64 bits @ 250 MHz | 64 bits @ 125 MHz<br>64 bits @ 250 MHz |
| 2 | 64 bits @ 62.5 MHz<br>64 bits @ 125 MHz<br>64 bits @ 250 MHz | 64 bits @ 125 MHz<br>64 bits @ 250 MHz | 64 bits @ 250 MHz<br>128 bits @ 125 MHz |
| 4 | 64 bits @ 125 MHz<br>64 bits @ 250 MHz | 128 bits @ 125 MHz<br>64 bits @ 250 MHz | 128 bits @ 250 MHz<br>256 bits @ 125 MHz |
| 8 | 128 bits @ 125 MHz<br>64 bits @ 250 MHz | 256 bits @ 125 MHz<br>128 bits @ 250 MHz | 256 bits @ 250 MHz |
| 16 | 128 bits @ 250 MHz | 256 bits @ 250 MHz | 512 bits @ 250 MHz |

The detector data generated by the particle detection are usually short. For example, for the STS-XYTER (also known as SMX) [4]), the data are 24 bits long. In the concentrated stream, the data must be accompanied by metadata describing its origin, resulting in a final size of 32 bits. Hence, the data concentrator must efficiently pack 32-bit detector data into 256-bit PCIe data.

The problem may be generalized as described in the next section.

### 1.1. Formulation of the Problem

The system receives the data words from $N$ inputs at frequency $f_{in}$, and puts them into the records able to store $M$ words, which are read at $f_{out}$ frequency. The system has sufficient bandwidth. The following condition is met:

$$N \cdot f_{in} \leq M \cdot f_{out}$$

The intensity of the data stream delivered by the inputs may be different and may vary in time. Therefore, each piece of input data is associated with the "valid" flag. If the input is idle in the current clock period or contains data that should not be transferred to DAQ, the flag is deasserted. Otherwise, it is asserted confirming that the particular input delivers "valid data". When the data are concentrated, the inputs with the deasserted "valid" flag should be skipped, not creating "holes" in the output stream. For event reconstruction in the triggerless DAQ, the data must be assigned to a particular time period (Of course, certain tolerance is unavoidable. Therefore, some overlap between consecutive analysis periods is used). Therefore, an essential requirement is that the concentrator disturbs the time-ordering of the input data as little as possible. The inputs should be scanned sequentially using a round-robin approach. A single data word should be copied to the lowest free location in the output record if valid data are found. After all $M$ positions are filled, the output record is sent to DAQ and cleared afterward (Even the incomplete (but properly marked) output record may need to be sent at the end of the analysis period. However, this mechanism remains outside the scope of this paper.).

In the next chapter, the existing solutions to the concentration problem are presented, and their disadvantages are discussed.

## 2. Existing Solutions for Concentrators

Unfortunately, finding published information about the data concentration methods used in existing FPGA-based data concentrators is difficult. The presented review is based on the few available publications and presentations, publicly available source code analysis, or the authors' experience.

Most existing solutions may be grouped into two categories described in the following sections.

## 2.1. High Speed Polling

The trivial solution is a direct implementation of the procedure described in Section 1.1, based on browsing all inputs at the frequency $f_{scan} = N \cdot f_{in}$ (see Figure 1).

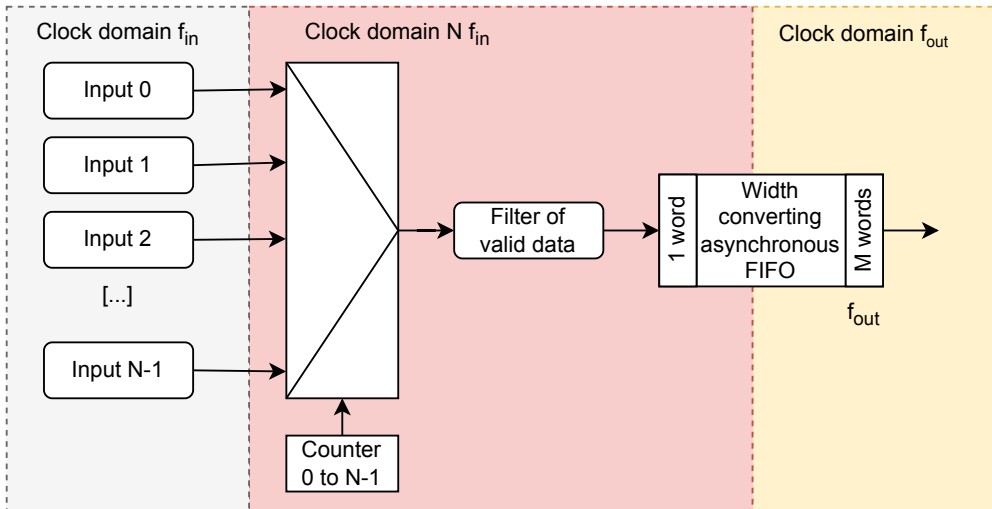

**Figure 1.** Structure of the concentrator based on high-speed polling. The central area must work with clock frequency $N \cdot f_{in}$, which may be too high for FPGA.

If valid data are found, it is copied to the asymmetric FIFO queue with an input width of one word and an output width of $M$ words. This warrants that no valid data are skipped, and the output words are filled with valid data only. The only problem with that solution is that the necessary $f_{scan}$ may be too high to be acceptable for FIFO in FPGA. Therefore, this method may be used only in case of low input clock frequency or concentrating data from a small number of inputs.

This solution is used for concentrating the data from STS-XYTER2 [4] front-end ASICs transmitted through GBT Links [5] in the readout chain of the STS detector in the CBM experiment. In that readout, each GBT Link transports data from 14 E-Links working with a 320 Mb/s rate. The 8b/10b encoded hit data occupy 30 bits in the E-Link. Therefore, the hit data rate in the individual E-Link is not higher than $\frac{320 \text{ Mb/s}}{30 \text{ bits}} \approx 10.67$ MHz. This is a low frequency. Because $14 \times 10.67$ MHz $< 160$ MHz, the data can be safely concentrated by consecutive scanning all 14 E-Link outputs at 160 MHz.

Another example is the firmware for the CRU board [6] used by the ALICE experiment at LHC at CERN. It uses a round-robin scanning of the output of FEE links (see [6], Figure 8). Then, only the valid data are packed into the 256-bit wide FIFO, delivering the data to PCIe. The authors do not describe at which frequency the inputs are scanned.

## 2.2. Width Conversion in Input Channels

High-speed scanning may be avoided if width conversion is performed in the input channels. In this solution, the small FIFOs with one-word wide input and $M$-words wide output are placed in each input channel, as shown in Figure 2.

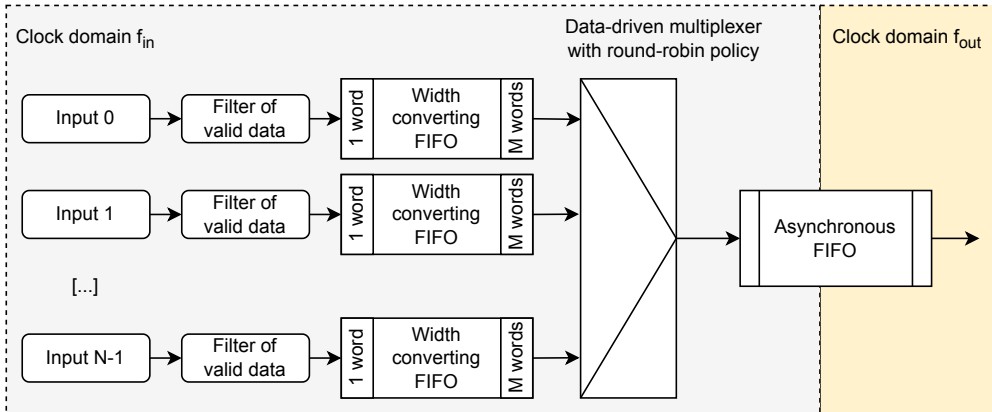

**Figure 2.** Structure of the concentrator with width conversion in each input link.

This solution does not require using a very high clock frequency. However, it has other disadvantages. If the link occupancy significantly differs between inputs, the concentration may significantly change the time-ordering of data. The data from low-rate links may get significantly delayed until $M$ data words are collected. This problem may be solved by introducing the timeout, after which the non-empty FIFO outputs its content even if it contains less than $M$ words. However, this modification results in inserting "holes" into the concentrated stream. Another disadvantage is the necessity to use a separate width-converting FIFO in each input channel. Those FIFOs may have limited depth, enabling implementation based on distributed RAM, but they may still increase resource consumption. Finally, this solution cannot use a simple counter-driven multiplexer periodically browsing the data. This solution requires a more complex data-driven multiplexer, which automatically selects the first input providing the complete data record after the previously serviced one (i.e., it implements the round-robin policy).

This approach seems to be used in the firmware for the FELIX board in the ATLAS experiment readout at CERN [7]. Unfortunately, the operation of the data concentrator in the FELIX firmware is not described in detail in any paper. However, the sources of that firmware are publicly available, enabling analysis of the concentrator code. The concentration is carried out in the CRToHostdm module [8] containing the asymmetric FIFO responsible for concatenating a few words and width conversion. Outputs of FIFOs from multiple channels are scanned in the CRToHost module [9].

*2.3. Need for Another Concentration Method*

None of the above methods matches all the requirements described in Section 1.1. Therefore, a new method is proposed in the next section.

**3. Proposed Solution—Concentration with the Direct Routing of Data**

It is possible to avoid the disadvantages of the previously described solutions by directly routing data from inputs to the proper position of the output record. Such a solution is shown in Figure 3.

The key functionality needed in this method is the capability to write the data from each individual input to a selected position in the output record. A dedicated controller calculates the desired location of data from each input. The controller must keep track of the occupancy of the output data. Additionally, it receives the "valid" flags from the input words. The controller starts with an output record occupancy equal to zero. If, for example, it receives valid words in three inputs, it routes them to positions 0, 1, and 2 in the output record and changes the occupancy to 3. The next valid word will be routed to position 4, and so on. When the output record is filled, the output strobe is generated, the collected words are transferred to the output FIFO, and the occupancy is set to 0.

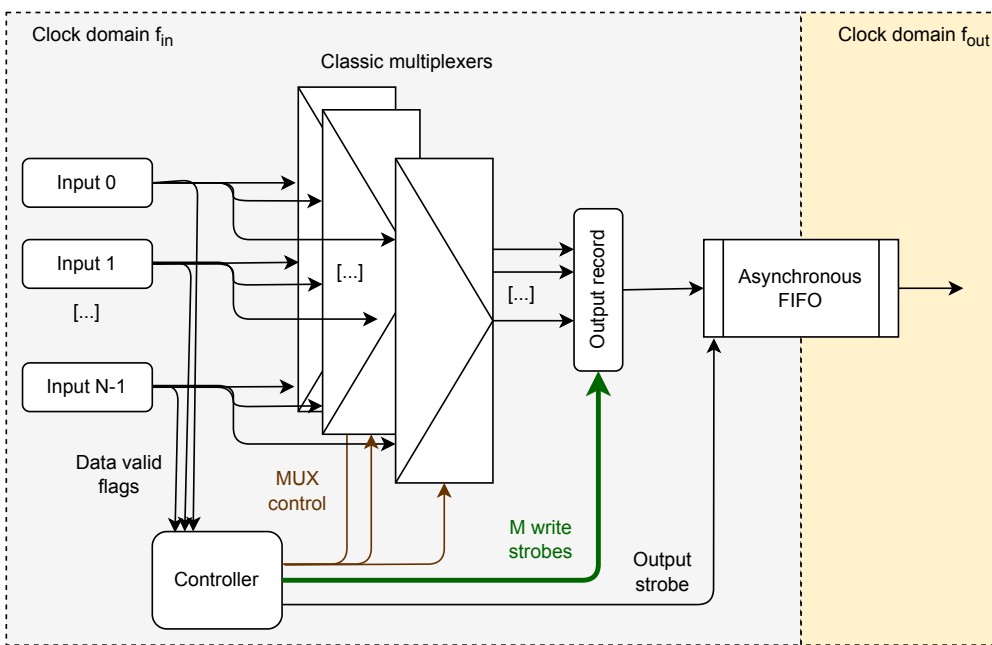

**Figure 3.** Structure of the concentrator with the direct routing of data. The controller keeps track of the current output record occupancy and routes each valid input word to the right position in the output record. When the word is completely filled, the output strobe is generated.

There is, however, a problem if the concentrator receives more valid data than needed to fill the output record. Those superfluous words must be stored somewhere. For that purpose, an "auxiliary record" register is introduced. The controller generates a write strobe for both registers. The output strobe causes the transfer of the output record to FIFO and, at the same time, of the auxiliary record to the output record. The modified concentrator is shown in Figure 4.

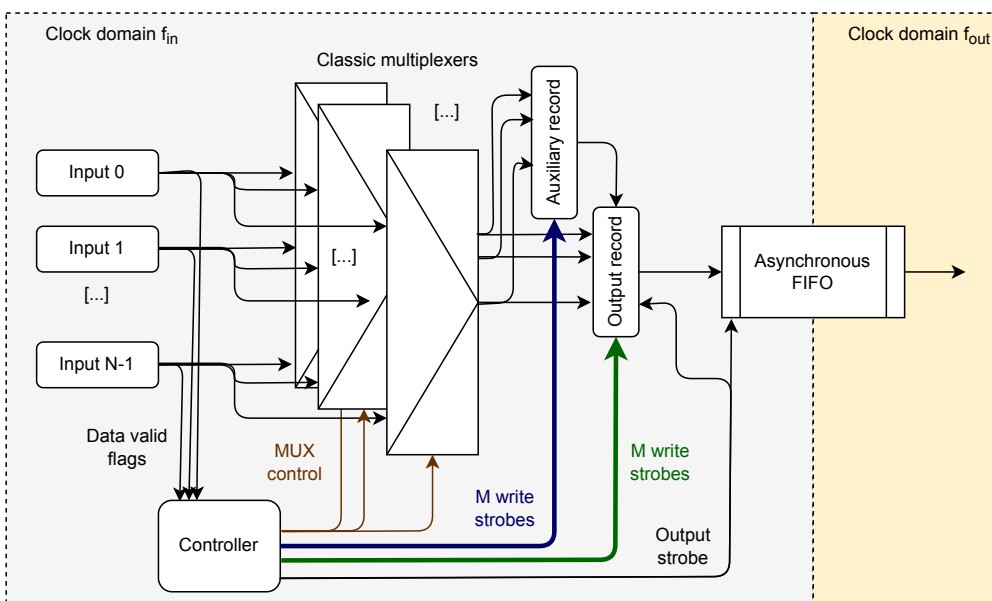

**Figure 4.** Structure of the concentrator with the direct routing of data and added auxiliary record. The controller keeps track of the current output record occupancy and routes each valid input word to the right position in the output record. When the word is completely filled, the output strobe is generated. If the number of valid words exceeds the number of empty positions in the output record, these extra words are stored in the auxiliary record. When the output strobe is generated, the content of this record is moved to the output record.

The presented concentrator should work correctly, but its implementation in FPGA is inefficient. Implementing $M$ multiplexers routing the data words consumes many resources and generates long critical paths in the FPGA. Therefore, yet another modification is needed. The multiplexers must be replaced with more efficient blocks for routing the data.

### 3.1. Concentrator Based on Beneš Network

A similar problem had to be solved in telecommunication networks for routing connections. Networks enabling arbitrary data permutation between their inputs and outputs are known as Beneš networks and have been described in [10]. An example of such network routing eight inputs to eight outputs is shown in Figure 5.

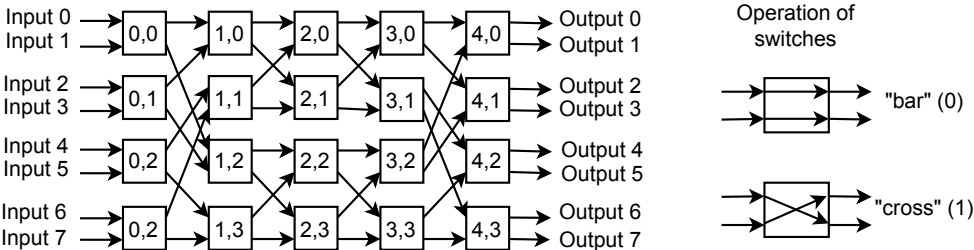

**Figure 5.** The Beneš network able to perform any permutation of eight inputs to eight outputs [11].

The Beneš network uses simple switches with two inputs and two outputs, transmitting the data transparently or swapping them. They may be efficiently implemented in FPGA. The lengths of all data paths are the same, so this network can be efficiently pipelined, which results in a short critical path. The general scheme of the concentrator based on the Beneš network is shown in Figure 6.

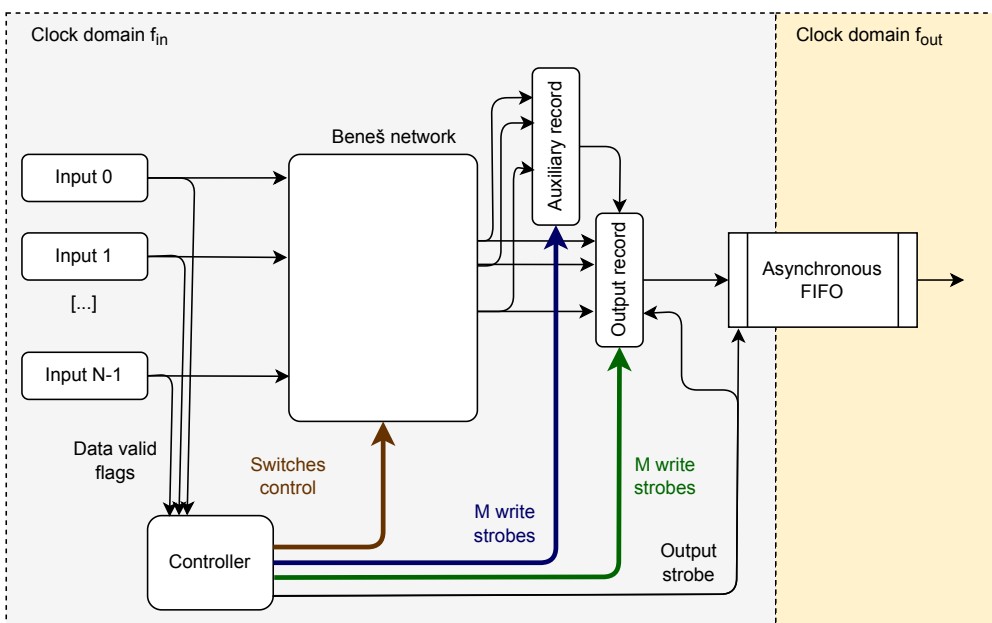

**Figure 6.** General structure of the concentrator based on the Beneš network.

The problem with the Beneš network is that its complexity quickly grows when the number of inputs and outputs increases. For example, the $4 \times 4$ Beneš network requires six switches in three layers, the $8 \times 8$ Beneš network—20 switches in five layers, and the $16 \times 16$ Beneš network—56 switches in seven layers. Generally, for $2^N$ inputs and outputs, the network requires $2^{N-1} \cdot (2 \cdot N - 1)$ switches in $2 \cdot N - 1$ layers.

Additionally, finding the configuration of switches that provides the required data routing is a complex task [11]. For small networks, it is possible to use a "brute force" approach

to check all possible configurations and create a table with configurations needed for all possible routings. For an $8 \times 8$ network, it is necessary to analyze $2^{20} = 1,048,576$ possibilities and find the right configuration for $8! = 40,320$ possible permutations. For a $16 \times 16$ network, the number of possible switch configurations is $2^{56} \approx 7.2 \times 10^{16}$, and the number of possible permutations is $16! \approx 2.1 \times 10^{13}$. Therefore, neither analysis of all possible configurations nor storing the right configuration for each possible permutation is viable. Therefore, an $8 \times 8$ network is used as a basis for the concentrator with the structure shown in Figure 7.

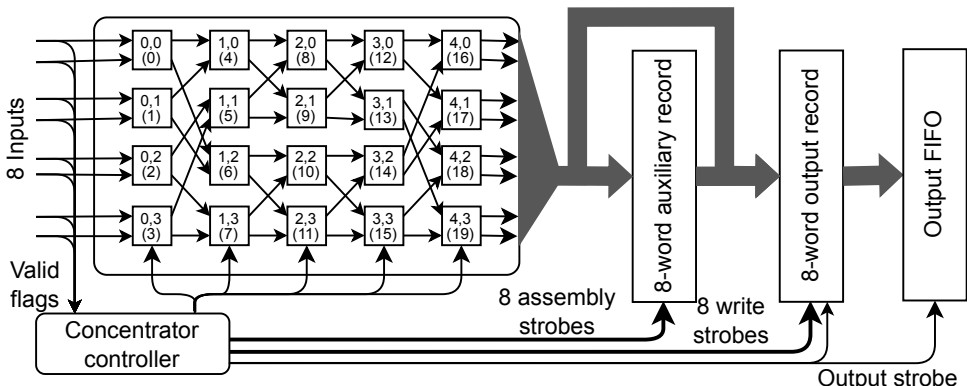

**Figure 7.** Data concentrator based on $8 \times 8$ Beneš network.

A simple C utility was written to investigate the switch settings providing different data permutations, as shown in Listing 1. It generates a simple file with lines containing the value of the switch configuration word and the data permutation it generates. This generated file is then read by the Python utility, which creates a dictionary where the key is the particular permutation, and the value is the smallest value of the switch configuration word that provides it.

The routing of data required by the HEP data concentration procedure described in Section 1.1 does not require all possible permutations. The concentrator always takes all valid data from the inputs and writes them without changing their order (in modulo $M$ sense) to the output record or the auxiliary record. The number of possible "valid" flag combinations is equal to $2^8 - 1$ (the trivial combination with all "valid" flags cleared is excluded). There are eight possible values of the first free position in the output record (0 to 7, depending on occupancy). Therefore, the number of required permutations equals $8 \times (2^8 - 1) = 2040$, much smaller than the number of all possible permutations ($8! = 40,320$). Such a vast reduction in the number of needed permutations should allow significant simplification of the data routing network. Finding the structure of the necessary simplified network was performed again using the "brute force" approach. The Python utility iterated over all eight possible occupancies of the output record and all possible combinations of the input data valid flags. The smallest values of the configuration word producing the necessary permutation were stored. It appeared that all so-produced configuration values were below $0 \times 400$. Only the lowest 11 bits controlling the switches in layers 0 to 2 were changing. The switches in layers 3 to 5 always transmitted the data without swapping. It enabled removing layers 3 to 5. The simplified network is shown in Figure 8.

**Listing 1.** C model of the 8 × 8 Beneš network.

```c
#include <stdio.h>
#include <stdlib.h>
#include <stdint.h>

typedef uint8_t t_data;

inline void swap(t_data * i1, t_data * i2,
                 t_data * o1, t_data * o2, uint32_t sw)
{
    if(sw) {
        *o1 = *i2;
        *o2 = *i1;
    } else {
        *o1 = *i1;
        *o2 = *i2;
    }
}

int main(int argc, char * argv[]) {
    uint32_t isw = 0;
    uint8_t sw[5][4];
    FILE * fout=fopen(argv[1],"wt");
    for (isw = 0; isw < (1<<20); isw++) {
        //Define the layers
        t_data l0[8],l1[8],l2[8],l3[8], l4[8], l5[8];
        //Initialize l0
        for(int i=0; i<8; i++) l0[i]=i;
        //1st layer of switches
        swap(&l0[0],&l0[1],&l1[0],&l1[4],isw & (1<<0));
        swap(&l0[2],&l0[3],&l1[1],&l1[5],isw & (1<<1));
        swap(&l0[4],&l0[5],&l1[2],&l1[6],isw & (1<<2));
        swap(&l0[6],&l0[7],&l1[3],&l1[7],isw & (1<<3));
        //2nd layer of switches
        swap(&l1[0],&l1[1],&l2[0],&l2[2],isw & (1<<4));
        swap(&l1[2],&l1[3],&l2[1],&l2[3],isw & (1<<5));
        swap(&l1[4],&l1[5],&l2[4],&l2[6],isw & (1<<6));
        swap(&l1[6],&l1[7],&l2[5],&l2[7],isw & (1<<7));
        //3rd layer of switches
        swap(&l2[0],&l2[1],&l3[0],&l3[2],isw & (1<<8));
        swap(&l2[2],&l2[3],&l3[1],&l3[3],isw & (1<<9));
        swap(&l2[4],&l2[5],&l3[4],&l3[6],isw & (1<<10));
        swap(&l2[6],&l2[7],&l3[5],&l3[7],isw & (1<<11));
        //4th layer of switches
        swap(&l3[0],&l3[1],&l4[0],&l4[2],isw & (1<<12));
        swap(&l3[2],&l3[3],&l4[4],&l4[6],isw & (1<<13));
        swap(&l3[4],&l3[5],&l4[1],&l4[3],isw & (1<<14));
        swap(&l3[6],&l3[7],&l4[5],&l4[7],isw & (1<<15));
        //5th layer of switches
        swap(&l4[0],&l4[1],&l5[0],&l5[1],isw & (1<<16));
        swap(&l4[2],&l4[3],&l5[2],&l5[3],isw & (1<<17));
        swap(&l4[4],&l4[5],&l5[4],&l5[5],isw & (1<<18));
        swap(&l4[6],&l4[7],&l5[6],&l5[7],isw & (1<<19));
        fprintf(fout,"%8.8x:%d%d%d%d%d%d%d%d \n ",isw,
                (int)l5[0],(int)l5[1],(int)l5[2],(int)l5[3],
                (int)l5[4],(int)l5[5],(int)l5[6],(int)l5[7]);
    }
    fclose(fout);
    return 0;
}
```

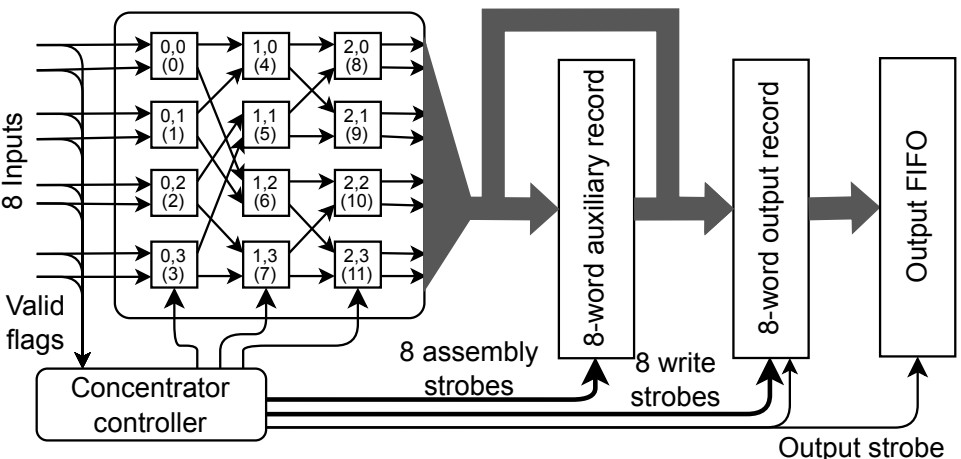

**Figure 8.** Data concentrator based on a reduced 8 × 8 Beneš network. Not all data permutations are needed to solve the concentration problem. Only three layers appeared to be sufficient for that purpose.

Removing the last three layers changed data routing from layer 2 to the output. Therefore, a new, reduced version of the C utility (see Listing 2) was necessary to find the correct value of the configuration word for each possible combination of input "valid" flags and output word occupancy.

**Listing 2.** C model of the reduced 8 × 8 Beneš network.

```c
#include <stdio.h>
#include <stdlib.h>
#include <stdint.h>

typedef uint8_t t_data;

inline void swap(t_data * i1, t_data * i2,
                 t_data * o1, t_data * o2, uint32_t sw)
{
    if(sw) {
        *o1 = *i2;
        *o2 = *i1;
    } else {
        *o1 = *i1;
        *o2 = *i2;
    }
}

int main(int argc, char * argv[]) {
    uint32_t isw = 0;
    FILE * fout=fopen(argv[1],"wt");
    for (isw = 0; isw < (1<<12); isw++) {
        //Define the layers
        t_data l0[8],l1[8],l2[8],l3[8], l4[8], l5[8];
        //Initialize them to "unknown" value: 9
        for(int i=0; i<8; i++)
            l0[8]=l1[i]=l2[i]=l3[i]=l4[i]=l5[i]=9;
        for(int i=0; i<8; i++) l0[i]=i;
        //1st layer of switches
        swap(&l0[0],&l0[1],&l1[0],&l1[4],isw & (1<<0));
        swap(&l0[2],&l0[3],&l1[1],&l1[5],isw & (1<<1));
        swap(&l0[4],&l0[5],&l1[2],&l1[6],isw & (1<<2));
        swap(&l0[6],&l0[7],&l1[3],&l1[7],isw & (1<<3));
        //2nd layer of switches
        swap(&l1[0],&l1[1],&l2[0],&l2[2],isw & (1<<4));
        swap(&l1[2],&l1[3],&l2[1],&l2[3],isw & (1<<5));
        swap(&l1[4],&l1[5],&l2[4],&l2[6],isw & (1<<6));
        swap(&l1[6],&l1[7],&l2[5],&l2[7],isw & (1<<7));
        //3rd layer of switches
        swap(&l2[0],&l2[1],&l3[0],&l3[4],isw & (1<<8));
        swap(&l2[2],&l2[3],&l3[2],&l3[6],isw & (1<<9));
        swap(&l2[4],&l2[5],&l3[1],&l3[5],isw & (1<<10));
        swap(&l2[6],&l2[7],&l3[3],&l3[7],isw & (1<<11));

        fprintf(fout,"%8.8x:%d%d%d%d%d%d%d%d \n",isw,
                (int)l3[0],(int)l3[1],(int)l3[2],(int)l3[3],
                (int)l3[4],(int)l3[5],(int)l3[6],(int)l3[7]);
    }
    fclose(fout);
    return 0;
}
```

### 3.2. Calculation of the Future Occupancy

The concentrator controller is also responsible for the calculation of the future occupancy of the output record. That value is needed in the next clock period. Therefore, its calculation is separated from finding the switch configuration. The switch configuration values may be stored in a BRAM-based look-up table. Adding pipeline registers before the Beneš network may compensate for the resulting delay. This is not possible in case of future occupancy. This must be calculated in combinational logic. The implementation used in the project is shown in Listing 3.

**Listing 3.** VHDL implementation of the combinational function calculating the future output record occupancy from the current one and the vector of valid input flags.

```vhdl
library ieee;
use ieee.std_logic_1164.all;
use ieee.numeric_std.all;
library work;

entity calculate_occupancy is
  generic (
    NOF_IN_WORDS : integer range 1 to 8 := 6);
  port (
    cur_occupancy  : in  integer range 0 to 7;
    valid          : in  std_logic_vector
                         (NOF_IN_WORDS-1 downto 0);
    next_occupancy : out integer range 0 to 7
    );
end entity calculate_occupancy;

architecture rtl of calculate_occupancy is
begin  -- architecture rtl
  calc : process (cur_occupancy, valid) is
    variable tmp : integer;
  begin  -- process calc
    tmp := 0;
    for i in 0 to NOF_IN_WORDS-1 loop
      if valid(i) = '1' then
        tmp := tmp+1;
      end if;
    end loop;  -- i
    tmp := tmp + cur_occupancy;
    if tmp > 7 then
      tmp := tmp - 8;
    end if;
    next_occupancy <= tmp;
  end process calc;
end architecture rtl;
```

## 4. Practical Implementations of the Concentrator

A Beneš-network-based concentrator appeared to be useful in different data acquisition systems currently developed. The design may be adjusted to particular needs, as shown in this section.

The solution based on $8 \times 8$ Beneš networks is needed for the GERI board [12] based on the Trenz TEC0330 PCIe card [13]. This board, when supplemented with an FMC card with 8 SFP+ cages (e.g., [14]), enables the concentration of data from 8 GBT Links to the DMA system [15] connected to the 8xGen 3 PCIe bus. The DMA system uses 256-bit data, which may be treated as a record containing eight 32-bit words. Thence, the solution described in the previous section may be directly applied.

If the GERI board is connected to the TFC system [16], one SFP+ cage is used for the TFC communication. In that case, a smaller $7 \times 8$ Beneš network is needed. It may be obtained from an $8 \times 8$ network. The 7th input should be connected to both inputs in the last switch in layer 0. This eliminates the need to control that switch. Its control input may be connected to a constant value. As a result, the number of switches to be controlled is reduced from 12 to 11. That configuration is shown in Figure 9.

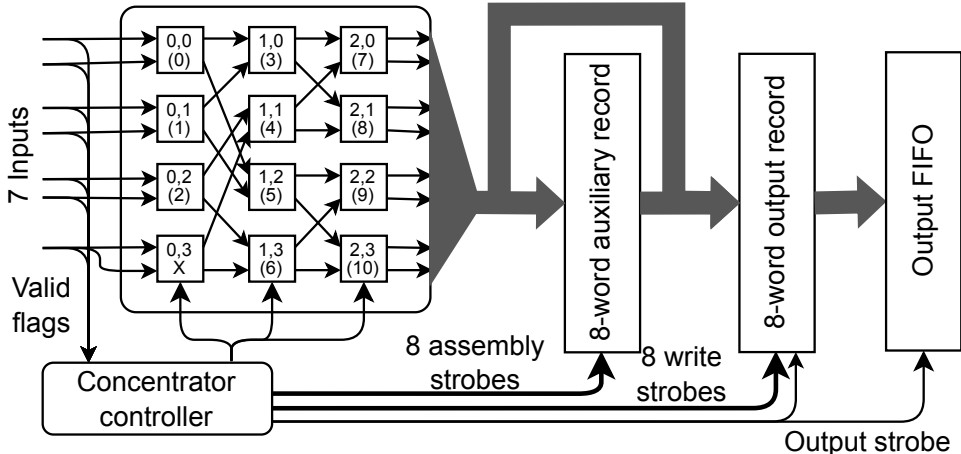

**Figure 9.** Data concentrator for seven inputs based on a Beneš network reduced to $7 \times 8$ size.

Special solutions are needed when it is necessary to concentrate data from more than eight inputs. Such a situation occurred in designing the firmware for the new CRI2 [17] readout board for the CBM [18] experiment. Currently planned hardware solutions [19] need to concentrate data either from 9 or 12 GBT Links delivering data at 160 MHz to a 256-bit wide word at a frequency up to 250 MHz. Of course, using the Beneš network with a size limited to $8 \times 8$ requires time multiplexing the input data. However, it does not require as high a frequency as the high-speed polling method described in Section 2.1. For these designs, a dedicated data converter has been developed, which receives two input data sets at frequency $f_{IN}$, combines them, and then outputs them as three smaller sets at the frequency $f_{OUT} = \frac{3}{2} f_{IN}$. In the described system, the Beneš network works at a frequency of 240 MHz, which is below 250 MHz.

In the case of 12 input links, this converter works with an $8 \times 8$ Beneš network, as shown in Figure 10.

In the case of nine input links, the converted data consist of three 6-word sets. They are delivered to the simplified $6 \times 8$ Beneš network, as shown in Figure 11.

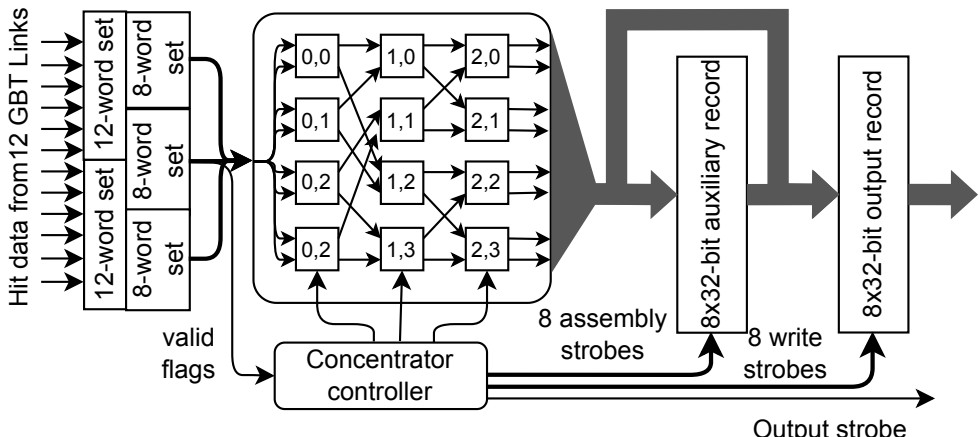

**Figure 10.** Data concentrator for 12 inputs. The data width converter receives input sets containing 12 words at 160 MHz, concatenates two such sets, and outputs them as three 8-word sets at 240 MHz. Further concentration is performed as in Figure 8.

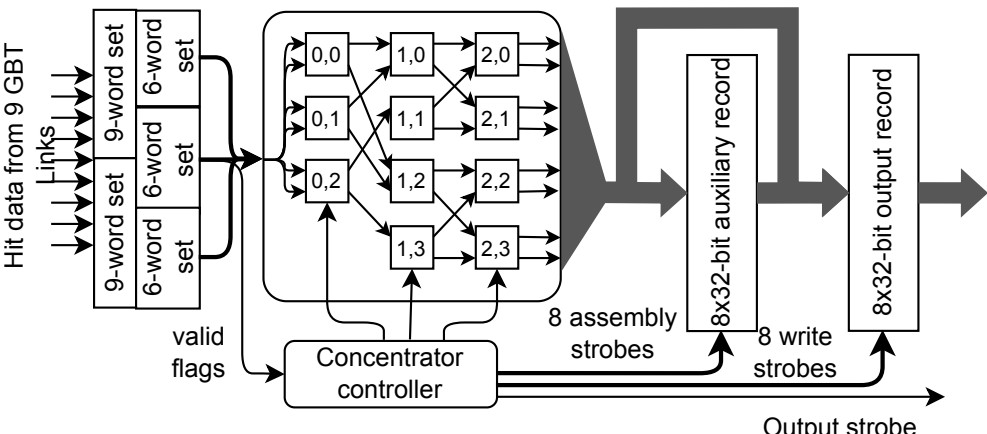

**Figure 11.** Data concentrator for nine inputs. The data width converter receives input sets containing nine words at 160 MHz, concatenates two such sets, and outputs them as three 6-word sets at 240 MHz. Further concentration is conducted via a Beneš network reduced to $6 \times 8$ size. The last two inputs and the fourth switch in layer 0 are removed. Only 11 switches are controlled, like in the case of a $7 \times 8$ network.

## 5. Implementation of the Concentrator in FPGA

The VHDL sources of the described concentrator are available in the repository [20]. This article describes the version tagged as "v1.0".

The type of concentrated data are defined in the **concentrator_pkg_p** package. It is set originally to a 32-bit standard logic vector but may be redefined by the user.

The **concentrator controller** block uses the table with precomputed configuration words for the Beneš network. It is provided by the **concentrator_lut_pkg** package generated automatically by utilities and scripts located in the **tools** subdirectory.

The top entity is **benes_concentrator** declared as shown in Listing 4.

The additional data-width adapter needed when the number of inputs must be increased to 9 or 12 is available as entity **converter_2to3** declared as shown in Listing 5.

Implementing the concentrator in pure VHDL facilitates its porting to FPGA chips of different vendors.

**Listing 4.** Declaration of the top entity of the Beneš network based data concentrator. The **NOF_IN_WORDS** parameter defines the number of inputs in the network (6, 7 or 8).

```vhdl
library ieee;
use ieee.std_logic_1164.all;
use ieee.numeric_std.all;

library work;
use work.concentrator_pkg_p.all;
use work.concentrator_lut_pkg.all;

entity benes_concentrator is
  generic (
    NOF_IN_WORDS : integer := 6);
  port (
    clk_i : in std_logic;
    rst_i : in std_logic;

    dav_i  : in std_logic_vector(NOF_IN_WORDS-1 downto 0);
    data_i : in t_conc_data_arr(NOF_IN_WORDS-1 downto 0);

    dav_o  : out std_logic;
    data_o : out t_conc_data_arr(8-1 downto 0)
    );
end benes_concentrator;
```

**Listing 5.** Declaration of the data width converter needed to concentrate 9 or 12 inputs. The **INSETSIZE** parameter defines the number of concentrated inputs. It must be 3/2 times bigger than the number of inputs of the connected Beneš network.

```vhdl
library ieee;
use ieee.std_logic_1164.all;
use ieee.numeric_std.all;

library work;
use work.concentrator_pkg_p.all;

entity converter_2to3 is

  generic (
    INSETSIZE : integer := 9);

  port (
    set_i     : in  t_conc_data_arr(INSETSIZE-1 downto 0);
    dav_i     : in  std_logic_vector(INSETSIZE-1 downto 0);
    in_clk_i  : in  std_logic;
    set_o     : out t_conc_data_arr((INSETSIZE*2/3)-1 downto 0)  := (others => (others => '0'));
    dav_o     : out std_logic_vector((INSETSIZE*2/3)-1 downto 0) := (others => '0');
    out_clk_i : in  std_logic;
    rst_p     : in  std_logic
    );

end entity converter_2to3;
```

## 6. Tests and Results

All five concentrator variants (with 6 to 9 or 12 inputs) have been tested in simulations. The input data were generated as 32-bit words containing consecutive numbers starting from $0 \times 100$. The user could set the probability that the data word is delivered to the individual input. The tests were performed for various values of probability: very low (0.01), low (0.1), medium (0.5), high (0.9), and very high (1.0). For all tested values of the probability, all five configurations of the concentrator worked correctly. All data delivered to the inputs were transmitted exactly once, and no invalid data were inserted into the

output records. The waveforms from a simulation of a 7-input concentrator at probability 0.5 is shown in Figure 12.

The most complex configuration with 12 inputs has also been verified in hardware. The implementation was performed in two boards:

- KCU105 [21] AMD/Xilinx board, equipped with Kintex Ultrascale XCKU040-2FFVA1156E FPGA;
- TEC0330 [13] board from Trenz Electronic equipped with Xilinx Virtex-7 XC7VX330T-2FFG1157C FPGA.

For testing in hardware, a special testbench has been prepared with the structure shown in Figure 13. The input data for the concentrator are written via PCIe to the FIFO with asymmetric port width (For 12 32-bit wide inputs, the necessary width of FIFO is 384 bits for data and 12 for valid flags, resulting in 396 bits. The Xilinx FIFO generator does not support that width. Therefore, a FIFO with 512-bit wide output was used. However, the minimal input width for such a FIFO is 64 bits. Therefore, the input value for that FIFO is concatenated from two PCIe-accessible registers. Writing one of them activates the FIFO write strobe.) Similarly, the output from the concentrator is written to the second FIFO with 256-bit wide input and 32-bit wide output connected to another PCIe-accessible register. Additional control and status registers support resetting the FIFOs and the concentrator, starting the data transfer, and reading the status of both FIFOs.

The design was successfully compiled for both selected platforms. Timing closure was obtained for 160 MHz and 240 MHz frequencies. The resulting resource consumption is shown in Table 2.

The FPGA configured with the testbench FIFO is controlled with the **uio_pci_generic** driver and a simple Python script. The script resets the FIFOs and the concentrator. Then, it prepares the input data sets and writes them to the first FIFO. Afterward, it starts the data transfer. Finally, it reads the concentrated data from the second FIFO. For automated tests, the input data sets are prepared similarly to the simulations. The data words containing consecutive values were written to the consecutive inputs. With the probability defined by the user, each input could be skipped.

The tests were repeated multiple times with different probabilities of skipping the input and a different number of input data sets (of course, always smaller than the capacity of the input FIFO). In all tests, the concentrated data were correctly delivered to the output FIFO without losses or duplications.

**Table 2.** Resource consumption of the 12-input data concentrator together with the testbench for chosen hardware platforms. Absolute and percentage (in parenthesis) consumption is given. The design was synthesized in the version where no BRAM was used for the controller. Separate values for the testbench, data width converter, and concentrator itself are given. That version uses the biggest concentrator based on the $8 \times 8$ Beneš network. For all other described configurations, the resource utilization will be lower.

| | KCU105 | | | TEC0330 | | |
|---|---|---|---|---|---|---|
| | **LUTs** | **Flip Flops** | **Block RAMs** | **LUTs** | **Flip Flops** | **Block RAMs** |
| Available | 242,400 | 484,800 | 600 | 204,000 | 408,000 | 750 |
| Whole testbench | 7685 (3.17%) | 10,641 (2.19%) | 36 (6.00%) | 10,406 (5.1%) | 12,339 (3.02%) | 36 (4.80%) |
| Data width converter | 745 (0.31%) | 1459 (0.30%) | 0 (0.0%) | 742 (0.36%) | 1459 (0.36%) | 0 (0.0%) |
| Data concentrator | 1038 (0.43%) | 1566 (0.32%) | 0 (0.0%) | 1032 (0.51%) | 1566 (0.38%) | 0 (0.0%) |

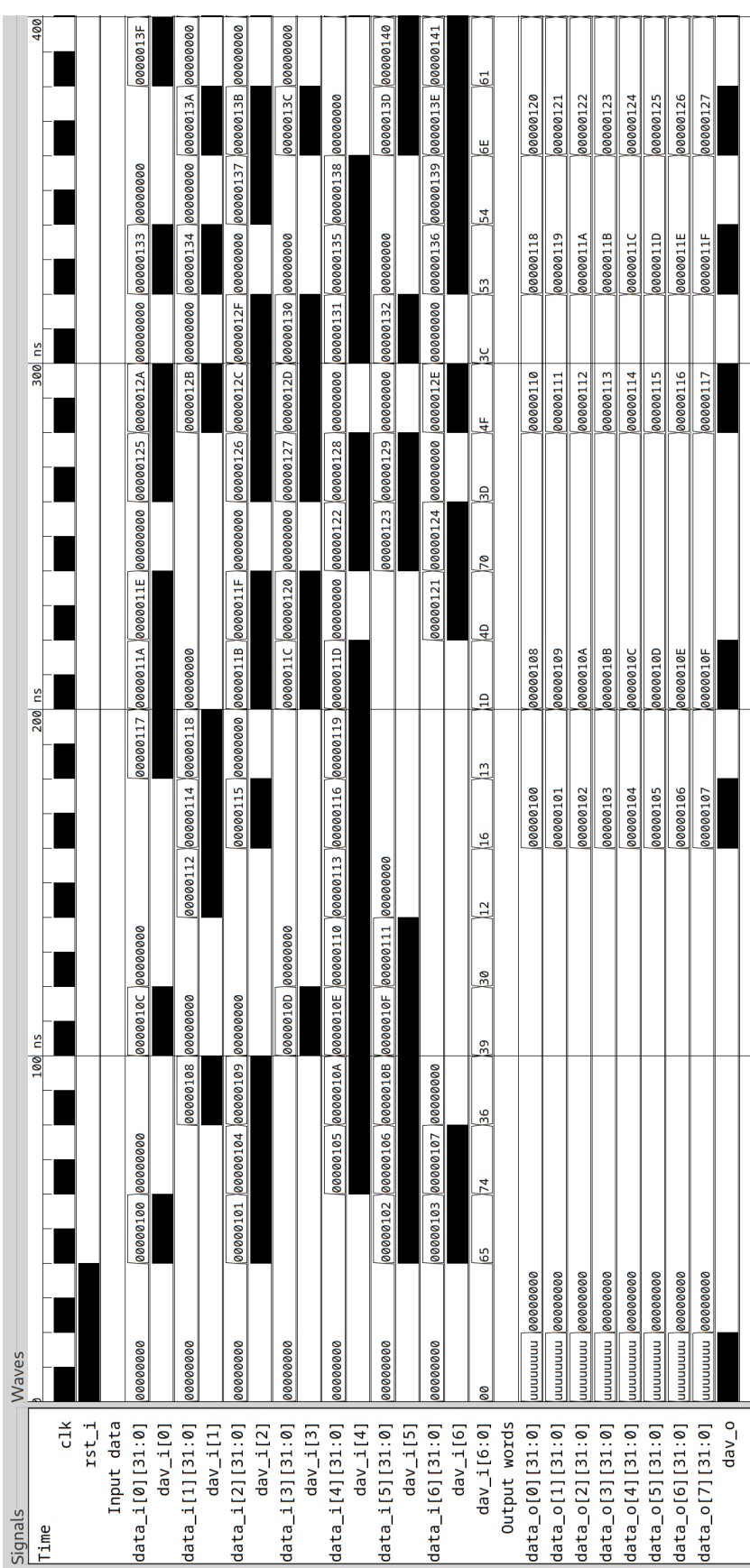

**Figure 12.** Results of simulation of the concentrator with seven inputs and probability of data presence set to 0.5.

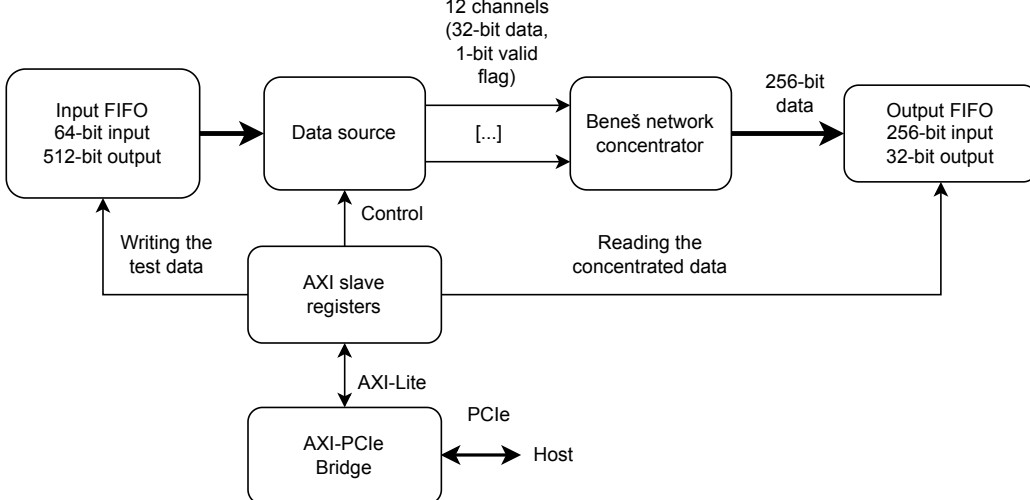

**Figure 13.** Testbench for testing the concentrator in the hardware.

## 7. Discussion

The proposed concentration method eliminates the disadvantages of the previously used solutions:

- The high-frequency polling (HFP) method described in Section 2.1 for $M$ inputs requires a local clock with frequency $M$ times bigger than the input clock.
  The proposed method for $M \leq 8$ does not require using a higher-speed local clock. For a higher number of inputs, a certain increase of local clock frequency is needed (see the 9 and 12 input versions described in Section 4). However, the multiplication factor is much lower than in the HFP-based solution.

- The method based on width conversion in the input channels (WCI) described in Section 2.2 for $M$ inputs requires $M$ width converting (asymmetric) FIFOs. It also needs a complex multiplexer that automatically finds the next valid data according to a round-robin policy. Additionally, this method may significantly delay the data from low-traffic inputs versus the data from high-traffic inputs, which may impair further data analysis.
  The proposed method does not use width-converting FIFOs. It also warrants that the data delivered in each input channel are written to the output or auxiliary record in the next clock cycle. In both methods, the prolonged lack of input data before the whole output record is filled may delay sending the concentrated data. A timeout mechanism is needed to prevent this. However, in the WCI method, incomplete records may be sent from all input channels, while in the proposed method, only one incomplete record will be generated;

- The method based on the direct routing of data via multiplexers (DRMX) described at the beginning of Section 3 is functionally equivalent to the proposed one. The differences are related to the implementation in the FPGA. For $M$ inputs, the DRMX method requires $M$ independent multiplexers. Even though modern FPGAs are equipped with dedicated multiplexer blocks, connecting all input channels to all multiplexers uses many routing resources. This may result in suboptimal routing and an increase in critical path latency.
  In the proposed method, each input channel is connected to the single input of one Beneš network switch. (For 9 or 12 inputs, this applies to each output from the width converter.) These switches are very simple (see Figure 5) and may include pipeline registers. Thence, routing these data in the FPGA may use few resources, and the latency of related critical paths is low.

The base of the proposed method is a well-known Beneš network technology [10]. However, it is used in a new application and optimized for its specific needs. The observa-

tion that the problem being solved requires only a subset of possible permutations enabled optimization of the network structure. The precalculation of the switch settings needed for each combination of "valid" flags and output record occupancy resulted in an efficient implementation of the concentrator controller.

## 8. Conclusions

This paper presented a new approach for high-speed concentrating of low-width data received from detectors' front-end electronics to high-width records transmitted via the PCIe interface to the DAQ computer. In comparison to the previously used concentration methods, the presented method gives the following benefits:

- It does not require a high-speed clock to scan the data to be concentrated;
- It uses low-complexity routing of data inside FPGA;
- Pipeline registers may be added after selected (or even all) routing stages;
- The concentrated data are delivered to the output in the original order (ordering the data in each concentrated stream is preserved, and multiplexing of streams is based on a round-robin approach);
- Even the data from low-traffic inputs are quickly delivered to the output record.

A significant benefit of the proposed solution is its open-source character. It is available in a public repository [20] under a permissive dual GPL/BSD license.

Future work may be focused on investigating possibilities of efficient concentration of data to the output record with a capacity of 16 words. Such a solution may be needed for new PCIe versions requiring a 512-bit-wide data bus in FPGA.

**Author Contributions:** Conceptualization, W.M.Z. and B.M.Z.; software, W.M.Z. and M.G.; validation M.G.; investigation, M.K.; writing—original draft preparation, W.M.Z.; writing—review and editing, all authors; supervision, W.M.Z.; specific technical contribution of the authors: general concept of the solution and software simulations, W.M.Z.; Concept of using the Beneš network for data routing in the concentrator, B.M.Z.; Implementation in HDL, simulations, and testing in hardware, M.G.; review of previous art, M.K.; the percentage contribution of the authors is: M.G.—40%, W.M.Z.—35%, M.K.—10% and B.M.Z.—15%. All authors have read and agreed to the published version of the manuscript.

**Funding:** This research was partially supported by the statutory funds of the Institute of Electronic Systems. This project has also received funding from the European Union's Horizon 2020 research and innovation programme under Grant No. 871072.

**Data Availability Statement:** Not applicable.

**Conflicts of Interest:** The authors declare no conflict of interest.

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
