# Peer review of "Beneš Network-Based Efficient Data Concentrator for Triggerless Data Acquisition Systems"

_electronics, doi:10.3390/electronics12061437_

Round 1
Reviewer 1 Report
Interesting study.
However, is the main contribution of the paper implementing a variant of Benes network for High-Energy Physics (HEP) Data Acquisition Systems (DAQs), since the concept itself is not entirely novel?
It also seems like special optimization was performed for these applications (page 7). It will be useful to the community to explain why and how to optimize the benes network for applications like the HEP DAQs.
Author Response
Dear Reviewer,
Thank you for your review. We have considered your remarks. We hope that has improved the article's quality.
Below are our responses:
Interesting study.
Thank you for that encouraging comment.
However, is the main contribution of the paper implementing a variant of Benes network for High-Energy Physics (HEP) Data Acquisition Systems (DAQs), since the concept itself is not entirely novel?
The old "Discussion and conclusions" section has been split into separate "Discussion" and "Conclusions" sections. The new "Discussion" part better explains the contribution of the paper.
It also seems like special optimization was performed for these applications (page 7). It will be useful to the community to explain why and how to optimize the benes network for applications like the HEP DAQs.
The ground for possible optimizations was described in lines 151-156 (in the first of the version). However, maybe it was not clear.
That part of section 3.1 has been rewritten to explain the applied optimizations better.
Reviewer 2 Report
This paper presents a good value of technical material. However, it needs improvement.
1- You have mentioned in the abstract:
At high and varying 2 data rates combined with the large width of the concentrator’s output interface, this task is non-trivial.
You should at least bring a reason why this is not trivial.
2- why this is well suited for FPGA. There is no provided reason.
3- The conclusion should be re written and drafted. You can take a look to other good samples in your field.
Author Response
Dear Reviewer,
Thank you for your review. We have considered your remarks. We hope that has improved the article's quality.
Below are our responses:
This paper presents a good value of technical material. However, it needs improvement.
Thank you for the encouraging comment. We have applied your remarks to improve the paper.
1- You have mentioned in the abstract:
At high and varying 2 data rates combined with the large width of the concentrator’s output interface, this task is non-trivial.
You should at least bring a reason why this is not trivial.
The abstract has been extended to explain difficulties related to dense high-speed data concentration. Of course, the description is short due to the limited length of the abstract.
2- why this is well suited for FPGA. There is no provided reason.
The abstract now lists the features that make the design well-suited for FPGA implementation. Of course, the description is short due to the limited length of the abstract.
3- The conclusion should be re written and drafted. You can take a look to other good samples in your field.
Indeed the conclusions were rather messy. Section "Discussion and conclusions" has been split into separate "Discussion" and "Conclusions" sections. The "Conclusions" have been rewritten in a better-organized way.
Reviewer 3 Report
The benefit of proposed concentrator is clearly describe in this paper. On the other hand, I have two questions as follows.
1. How to implement it on FPGA? Please show the design flow.
2. The benefit is qualitatively described in section 3, on the other hand there is no quantitative comparison with other works.
Author Response
Dear Reviewer,
Thank you for your review. We have considered your remarks. We hope that has improved the article's quality.
Below are our responses:
The benefit of proposed concentrator is clearly describe in this paper. On the other hand, I have two questions as follows.
Thank you for the encouraging comment. We have answered your questions and suggestions below.
1. How to implement it on FPGA? Please show the design flow.
The new "Implementation of the concentrator in FPGA" section has been added (including listings 4 and 5).
2. The benefit is qualitatively described in section 3, on the other hand there is no quantitative comparison with other works.
A detailed comparison with the previously available solutions is provided in the "Discussion" section.